# White Phosphate Coatings Obtained on Steel from Modified Cold Phosphating Solutions

Evgeniy Rumyantsev, Varvara Rumyantseva and Viktoriya Konovalova *

Institute of Information Technology, Natural Sciences and Humanities, Ivanovo State Polytechnic University, 153000 Ivanovo, Russia; naturer@yandex.ru (E.R.); varrym@gmail.com (V.R.)
* Correspondence: kotprotiv@yandex.ru

**Abstract:** The article presents a method for obtaining white phosphate coatings on steel by cold method. The deposition of protective phosphate coatings was carried out from solutions based on the preparation "Majef", consisting of manganese and iron phosphates. To obtain phosphate films of white color, it is proposed to introduce zinc and calcium nitrates into phosphating solutions at the rate of 25–30 g/L. The surface of phosphate coatings was studied using the SolverP47-PRO atomic force microscope images, and the average grain size was determined. The structural and phase composition of phosphate coatings was been studied using X-ray diffraction analysis. The protective properties of phosphate coatings were estimated by corrosion rate indicators calculated from corrosion diagrams. Fine-crystalline uniform coatings were obtained from modified phosphating solutions at room temperature on steel. The white color of phosphate coatings is due to the increased content of phosphophyllite, hopeite, and parascholzite in their structural and phase composition. By applying protective phosphate coatings of white color on a steel product, corrosion can be slowed down by 4–4.5 times. However, white phosphate coatings are inferior in protective properties to unpainted coatings. The index of change in the mass of samples with white phosphate coatings because of corrosion is 0.371–0.41 g/(m²·h), and with unpainted coatings is 0.128 g/(m²·h).

**Keywords:** phosphate coating; cold phosphating; protection of steel; protective coating; corrosion protection

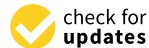



## 1. Introduction

The most used method of corrosion protection involves mass alloying or surface modification. However, surface modification is much more economical than mass alloying, and is more widely practiced. Methods commonly used for surface modification include the formation of a physical barrier to protect the metal from an aggressive environment [1–4].

Phosphating has long been successfully used as a method of protecting metal products and structures from corrosion [5–7]. The main advantage of the phosphate film is that it has high corrosion resistance in all types of combustible, lubricating and organic oils, in benzene, toluene and in all gases except hydrogen sulfide [8,9]. Phosphate coatings are used to protect against corrosion in combination with paint and polymer films, oils, and waxes, facilitate cold deformation of metal, reduce the coefficient of friction, create an insulating layer in electrical engineering, etc., [5–12].

Priority areas for improving phosphating processes are improving the protective and other functional properties of coatings, reducing the concentration of solutions, temperature and processing time, simplifying adjustments, unifying phosphating compositions, and reducing the environmental hazard of processes [5,6,13–15].

All conventional phosphating solutions are based on dilute phosphoric acid solutions based on alkaline or heavy metal ions, which mainly contain free phosphoric acid and primary phosphates of metal ions contained in the bath [10,16]. Traditionally, phosphating is carried out by immersion of products in a bath with a phosphate solution

heated to 80–95 °C [5,6,16,17]. The duration of the phosphating process is 40–60 min and depends on the concentration of phosphate ions in the working solution and the processing temperature [16,17]. To accelerate the phosphating process, reduce the release of hydrogen and the oxidation of $Fe^{2+}$ to $Fe^{3+}$, oxidants $Zn(NO_3)_2$, NaF, $NaNO_2$ are introduced into the bath [18,19]. Phosphating in hot solutions consumes a large amount of energy. In addition, the main reaction of hydrolytic decomposition of phosphating preparations also occurs in the absence of coated parts in the bath, which leads to an increase in the free acidity of the solution and, accordingly, to an increase in the porosity of the films and a decrease in protective properties [5,6,16,20].

Phosphating at room temperature significantly reduces the reaction rate of hydrolytic decomposition of phosphating preparations towards the formation of free orthophosphoric acid, which almost eliminates the dissolution of crystal phosphates formed on the surface of the part during its phosphating. The most favorable temperature range of phosphating solutions is 20–30 °C [21,22]. However, the rate of formation of the phosphate film and its thickness in cold solutions is significantly reduced. To accelerate the process of interaction of steel with free orthophosphoric acid, oxidizing agents (nitrates, nitrites, fluorides) are introduced into the phosphating solution [22]. A very important advantage of cold phosphating methods is the wide possibility of controlling the process and using them for processing large surfaces and bulky complex structures by pulverizing the solution.

The introduction of additional metal salts into phosphating solutions can change the color of the deposited coating [17–19]. Phosphating compositions containing dyes have been developed for the deposition of blue and green phosphate films [23].

Phosphate coatings applied to steel, zinc, zinc-coated steel, aluminum, and other similar metals have a crystal structure with a crystal size from several to about 100 μm [24–27]. The composition of phosphate coatings includes many different components [28,29]. There are more than 30 phosphate compounds [19] found in the phosphate coating. The phase components included in the crystalline phosphate film have a different color, which is also reflected in the color of the film.

Usually, phosphate coatings are gray in color. The purpose of this work was to obtain white phosphate coatings on steel by cold method and study their protective properties.

## 2. Materials and Methods

### 2.1. Materials

Coatings from phosphate solutions were deposited on samples of steel grade St3 (the chemical composition of steel is shown in Table 1). The surface of the samples was pre-sanded with sanding paper, degreased in 3–5% $Na_2CO_3$ solution with the addition of solid soap (1–2%) at a temperature of 50–70 °C for 10 min, and etched in a 5% solution of sulfuric acid for 30 s.

**Table 1.** Chemical composition of steel samples.

| Components | C | Si | Mn | Ni | S | P | Cr | N | Cu | As | Fe |
|---|---|---|---|---|---|---|---|---|---|---|---|
| Ammount (%) | 0.14–0.22 | 0.15–0.3 | 0.4–0.65 | up to 0.3 | up to 0.05 | up to 0.04 | up to 0.3 | up to 0.008 | up to 0.3 | up to 0.08 | ~97 |

To obtain phosphate coatings of white color, it is proposed to additionally introduce zinc and calcium nitrates at the rate of 25–30 g/L into phosphating solutions of the composition, g/L: preparation "Majef"–35–45, $Zn(NO_3)_2$–50–65, $NaNO_2$–3–4, glycerin–1–2, trilon B–6–8, preparation OS-20–5–10 [30].

The drug "Majef" is a fine crystalline powder in which the proportion of phosphoric acid, in terms of $P_2O_5$ is 46–52%, the mass fraction of manganese Mn is not less than 14%, iron is not more than 0.5%, insoluble substances in water are not more than 6%, sulfates are not more than 0.7%, total acidity is not less than 25%.

The phosphating solution was prepared according to the following technology: the preparation "Majef" was dissolved in 500 mL of water heated to 60–80 °C, then the solution

was cooled to 20–30 °C; the remaining components and water were added to it to a volume of 1 L, and mixed until completely dissolved. Samples made of steel grade St3 were immersed in a bath with a phosphating solution and kept in it for 20–25 min at room temperature. After that, the samples were washed with running water and dried in air for 24 h.

### 2.2. Method of Surface Investigation Using an Atomic Force Microscope

A scanning atomic force microscope SolverP47-PRO (NT-MDT, Moscow, Russia) was used to analyze the surface of the samples under study. This is a universal device for complex studies of various objects with high resolution. This microscope allows researchers to examine the surface of samples in areas up to 50 μm × 50 μm in size. To determine the average grain size of the deposited phosphate coatings, a grid was constructed on the intermediate divisions on the axes in the images. The grain diameter was measured at the intersections of the lines.

### 2.3. X-ray Analysis of the Phase Composition of Phosphate Coatings

Diffractograms were taken on a powder X-ray diffractometer D8 Advance (Bruker AXS Gmbh, Karlsruhe, Germany) at an X-ray wavelength λ = 1.5405 Å. The essence of qualitative radiographic analysis is to compare experimentally determined values of interplane distances, lines, and their intensity with reference diffractograms.

### 2.4. Investigation of the Corrosion Rate of Steel by the Method of Contact Corrosion

The model corrosion element (Figure 1) was composed of a sample of steel St3 and a graphite electrode, which were placed in different elbows of a U-shaped vessel. The working surface of the steel sample was previously measured. The U-shaped vessel was filled with a 3% NaCl solution. A saturated silver chloride electrode was used as the reference electrode.

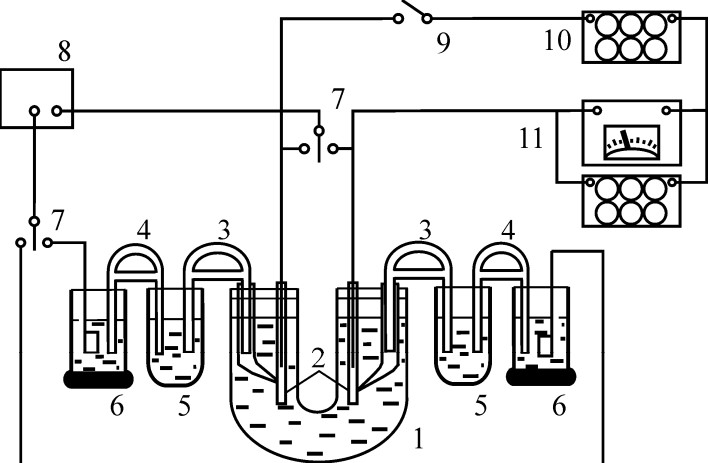

**Figure 1.** Installation for studying contact corrosion of metals: 1—U-shaped vessel; 2—test samples; 3—electrolytic keys with the test solution; 4—electrolytic keys with saturated KCl solution; 5—intermediate glasses with the test solution; 6—saturated silver chloride electrodes; 7—switches; 8—potentiometer; 9—knife switch; 10—resistance box; 11—microammeter with a shunt resistance box.

Using a digital voltmeter, the potentials of the electrodes were measured, and their polarity was determined with an open circuit. Then, the resistance at the decadal store was set: 50,000, 10,000, 5000, 1000, 500 and 100 Ω, and the potential values of the anode, cathode, and current were measured for each value of the external resistance.

For the highest value of the achieved current, corrosion rate indicators are calculated [31]:

$$K_m^- = \frac{j \times A}{z \times 26.8},\tag{1}$$

$$K_n = K_m^- \times \frac{8.76}{\rho_{me}},\tag{2}$$

where: $K_m^-$ is the rate of change of the sample mass, [g/(m²·h)]; $j$ is corrosion current density, [A/m²]; $A$ is atomic weight of metal, [g/mol]; $z$ is the valence of the metal; $K_n$ is rate depth of corrosion, [mm/year]; $\rho_{me}$ is density of metal, [g/cm³].

## 3. Results

As a result of cold deposition of modified phosphate coatings, uniform fine-crystalline matte gray coatings are formed (Figure 2a). Uniform fine-crystalline white coatings are obtained from a phosphating solution with an additional addition of zinc nitrate (Figure 2b), and from a phosphating solution with an addition of calcium nitrate; coatings are not deposited uniformly; whitish streaks and crystalline agglomerates are observed (Figure 2c).

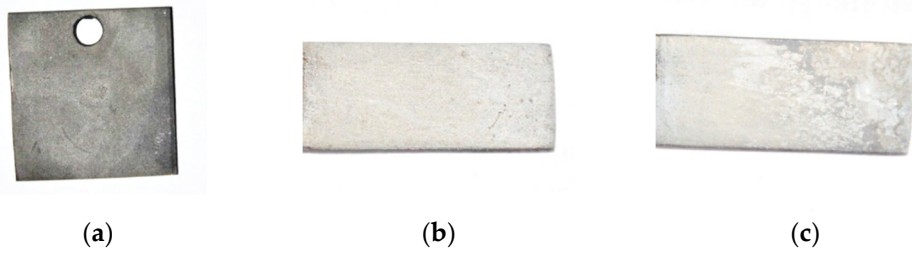

(**a**) (**b**) (**c**)

**Figure 2.** Images of the surface of phosphated steel samples obtained by a digital camera: (**a**) Unpainted phosphate coating from a modified solution; (**b**) White phosphate coating from a solution with the addition of zinc nitrate; (**c**) White phosphate coating from a solution with the addition of calcium nitrate.

### 3.1. Investigation of the Surface of Phosphate Coatings

According to the images from the atomic force microscope (Figure 3), it was found that the average grain diameter of films obtained from a modified solution is 165 nm, and for white phosphate coatings this parameter is 207 nm (coating from a solution with a high content of zinc nitrate) and 189 nm (coating from a solution with calcium nitrate).

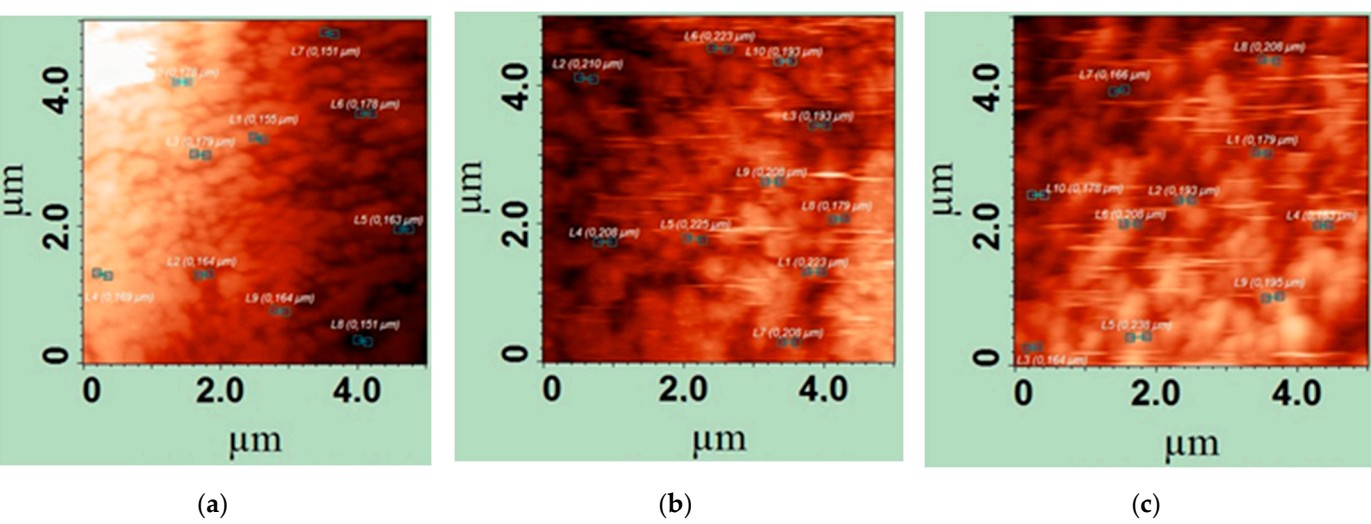

(**a**) (**b**) (**c**)

**Figure 3.** Images of the surface of phosphate coatings from the SolverP47-PRO atomic force microscope:

(**a**) Coating from a modified solution; (**b**) Coating from a solution with the addition of zinc nitrate; (**c**) Coating from a solution with the addition of calcium nitrate.

On 3-D models (Figure 4) of the surface, it can be seen that white phosphate coatings have a less-even surface relief compared to the modified phosphate coating. However, the addition of calcium and zinc salts to phosphating solutions reduces the surface roughness of deposited phosphate coatings, smooths out irregularities. The surface height decreased from 500 nm for modified coatings to 350 and 450 nm for a white coating with zinc nitrate and a white coating with calcium nitrate, respectively.

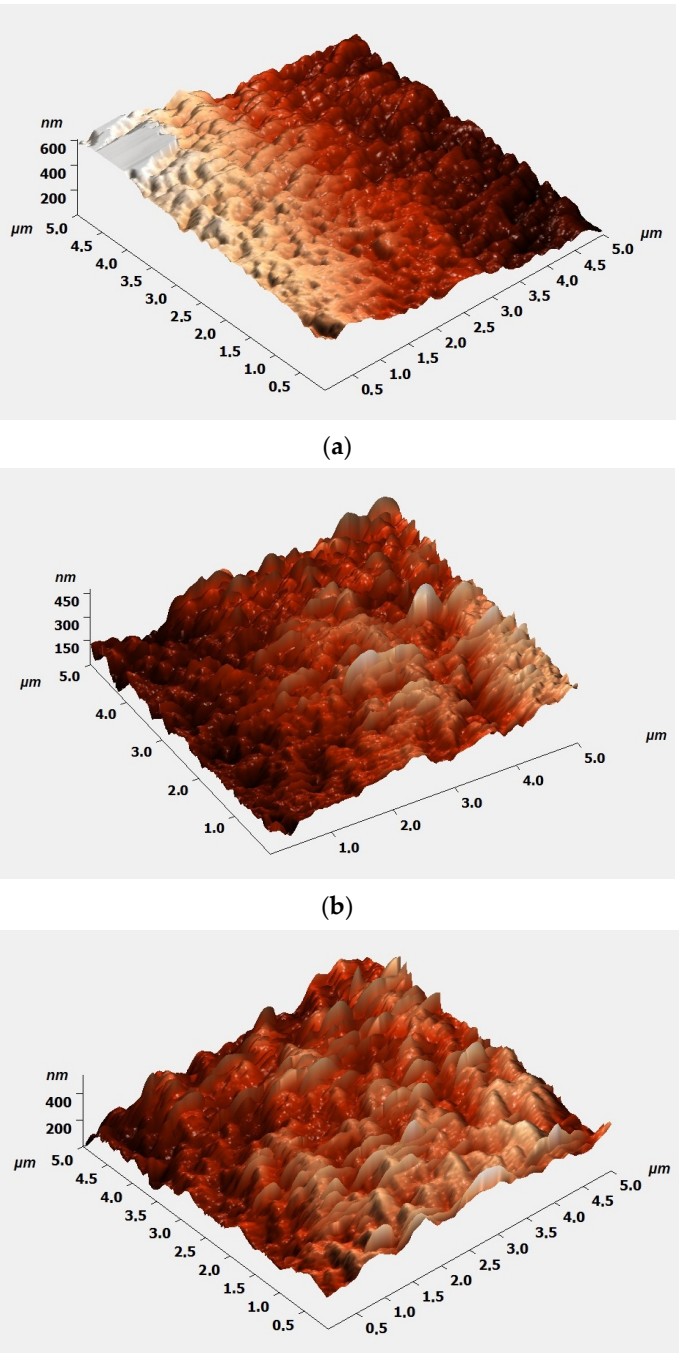

**Figure 4.** 3-D surface models of phosphate coatings: (**a**) From a modified solution; (**b**) From a solution with the addition of zinc nitrate; (**c**) From a solution with the addition of calcium nitrate.

### 3.2. Corrosion Diagrams Obtained by the Contact Corrosion Method

To calculate the speed and characteristics of the metal corrosion process, corrosion diagrams (Figures 5–8) are constructed graphically: $\varphi_a$–$i_a$—the curve of anodic polarization of the anode areas of the metal surface and $\varphi_k$–$i_k$—the curve of cathodic polarization of the cathodic areas of the metal surface. The analysis of the constructed corrosion diagrams consists of determining the controlling process, i.e., the stage of the process of electrochemical corrosion of the metal, which has the greatest resistance compared to other stages.

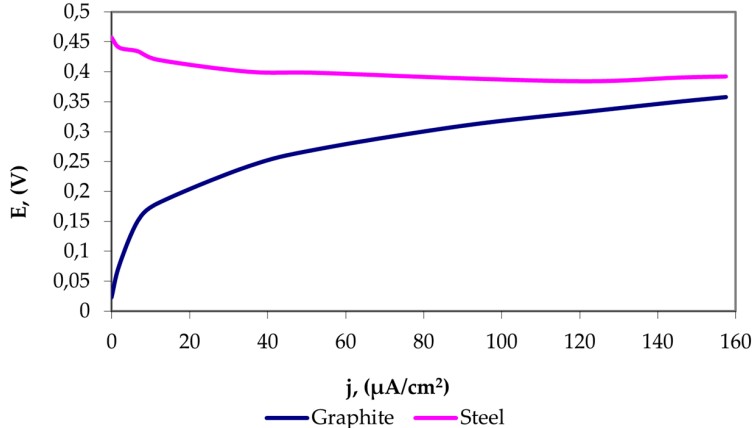

**Figure 5.** Corrosion diagram of a steel sample without protective coating.

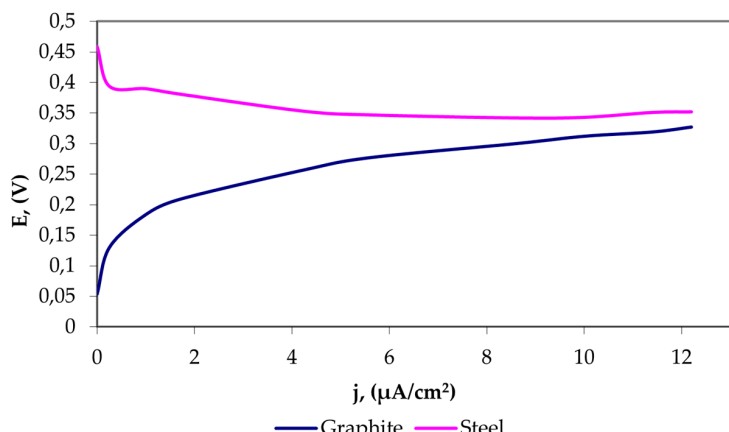

**Figure 6.** Corrosion diagram of a steel sample with an unpainted phosphate coating.

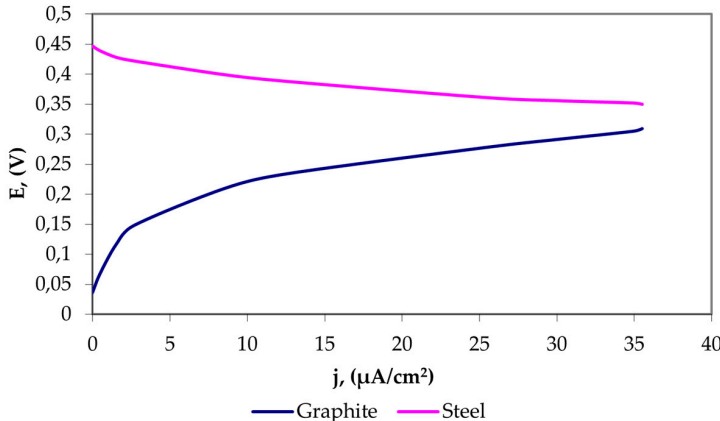

**Figure 7.** Corrosion diagram of a steel sample with phosphate coating from a solution with the addition of zinc nitrate.

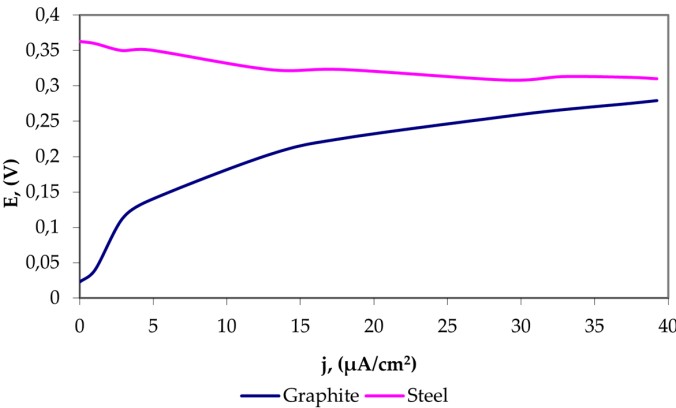

**Figure 8.** Corrosion diagram of a steel sample with phosphate coating from a solution with the addition of calcium nitrate.

### 3.3. Results of X-ray Analysis of the Phase Composition of Phosphate Coatings

Figures 9–11 show diffractograms of phosphate coatings characterizing their structural and phase composition. Analysis of diffractograms (Figures 9–11) showed that phosphate films obtained on steel from a modified solution based on the preparation "Majef" consist of several groups of crystals corresponding in phase composition to minerals: phosphoferrite $(Fe,Mn)_3(PO_4)_2 \cdot 3H_2O$ (#), phosphophyllite $Zn_2Fe(PO_4)_2 \cdot 4H_2O$ (+), reddingite $(Mn,Fe)_3(PO_4)_2 \cdot 3H_2O$ (●), hopeite $Zn_3(PO_4)_2 \cdot 4H_2O$ (*), switzerite $Mn_3(PO_4)_2 \cdot 7H_2O$ (□), vivianite $Fe_3(PO_4)_2 \cdot 8H_2O$ (○), strengite $FePO_4 \cdot 2H_2O$ (△).

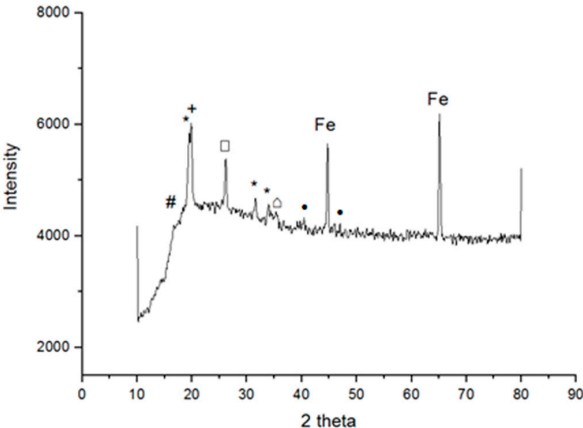

**Figure 9.** Diffractogram of unpainted phosphate coating.

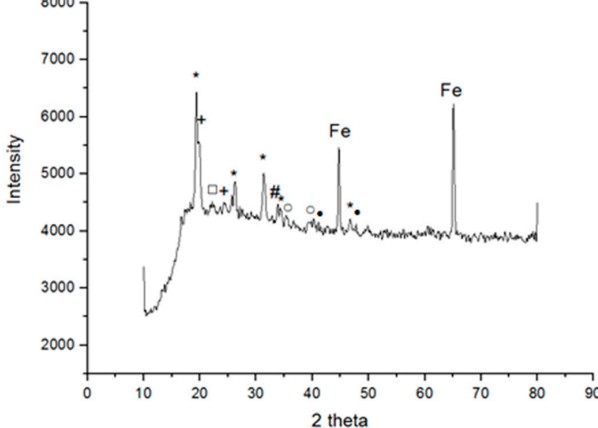

**Figure 10.** Diffractogram of a white coating from a phosphating solution with the addition of zinc nitrate.

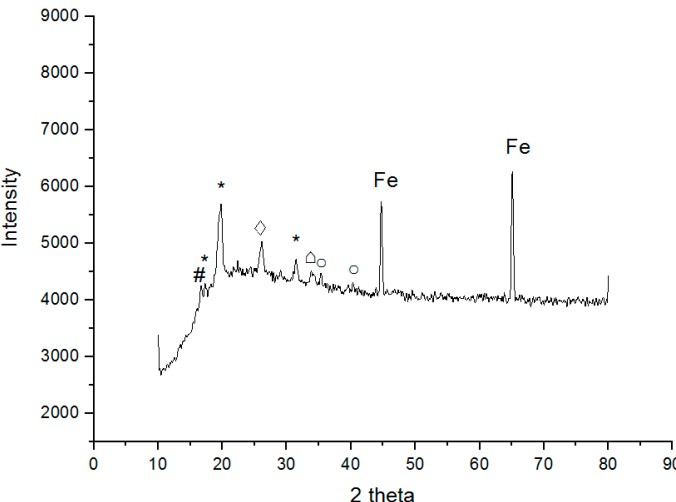

**Figure 11.** Diffractogram of a white coating from a phosphating solution with the addition of calcium nitrate.

## 4. Discussion

As can be seen from the images of the samples (Figure 2), the proposed additives of zinc and calcium nitrates in phosphating solutions change the color of the deposited coating. However, on the surface of the white phosphate coating, defective areas are formed from a solution with the addition of calcium nitrate. This spoils the appearance of the product, and such a phosphate coating cannot be used as a decorative one. Images of the surface of phosphate coatings obtained with an atomic force microscope (Figures 3 and 4) confirm the worse quality and greater surface roughness of white coatings from a solution with calcium nitrate addition compared with coatings from a solution with zinc nitrate addition. In this regard, it is advisable to further optimize the composition and mode of operation of the phosphating solution with the addition of calcium nitrate to obtain white coatings of good quality.

In white coatings obtained from phosphating solutions with the addition of zinc nitrate, the content of phosphophyllite and hopeite is increased. Crystals corresponding in phase composition to these minerals form the basis of such coatings. The mineral phosphophyllite is transparent crystals of colorless or light green color. Hopeite has a white color, apparently, it determines the color of the phosphate coating.

In white coatings from solutions with the addition of calcium nitrate, the content of vivianite is increased, the mineral parashcolzite $Zn_2Ca(PO_4)_2 \cdot 2H_2O$ ($\Diamond$) is also found. Vivianite is colorless, and changes color to blue-green in the air. Parashcolzite is transparent and has a white color.

Thus, by adjusting the content of a certain mineral in the structural-phase composition of the phosphate coating by changing the composition of the phosphating solution, it is possible to change the color of the deposited films.

The data obtained from corrosion diagrams (Figures 5–8) allowed us to calculate the corrosion rate of steel samples according to the Formulas (1) and (2). The data of tests of anticorrosive protection of coatings by contact corrosion (Table 2) show that steel samples protected by modified coatings corrode 10 times slower. White phosphate coatings reduce the corrosion rate by 4–4.5 times.

**Table 2.** Corrosion rates of steel samples with various modified phosphate coatings in a 3% NaCl solution.

| Corrosion Rate Indicator | Type of Coating | | | |
|---|---|---|---|---|
| | Without Coating | Unpainted Coating | White Coating Obtained from a Solution with Zinc Nitrate | White Coating Obtained from a Solution with Calcium Nitrate |
| Indicator of the change in the mass of the sample, $K_m^-$ (g/m$^2$·h) | 1.646 | 0.128 | 0.371 | 0.41 |
| Corrosion depth indicator, $K_n$ (mm/year) | 1.831 | 0.142 | 0.413 | 0.456 |

Since the smaller grain size causes higher performance properties of coatings [32–35], it is obvious that the introduction of zinc or calcium nitrate additives into phosphating solutions adversely affects the protective properties of the deposited phosphate films. Fine-crystalline phosphate coatings have high adhesion to the metal surface and provide an effective physical barrier to protect corroded metal products from the environment. Fine-crystalline coatings have less porosity and fewer cracks and voids in their structure [36–39]. Due to their insulating nature, phosphate coatings prevent the occurrence and spread of corrosion.

White phosphate coatings are worse at preventing the anodic dissolution of the metal compared to coatings obtained from modified phosphating solutions. The corrosion rate of St3 steel samples protected by unpainted modified phosphate coatings in 3% NaCl solution is 13 times lower, and in white phosphate films 4–4.5 times lower than in unprotected steel samples. The deep corrosion index of white phosphate coatings does not exceed 0.5 mm/year, and the permissible one is in the range of 0.3–0.5 mm/year. Thus, white phosphate coatings can be used as protective and decorative coatings for application on the surface of steel products.

**5. Conclusions**

1.  Solutions for the deposition of white phosphate coatings have been developed based on modified solutions of cold phosphating of steel. To make the phosphate coating white, it is proposed to introduce zinc or calcium nitrates into phosphating solutions at the rate of 25–30 g/L.
2.  Studies of the composition and structure of the resulting color coatings have been carried out. The grain size of white phosphate coatings is larger than that of unpainted modified phosphate coatings. However, the addition of calcium and zinc nitrates to phosphating solutions reduces the surface roughness of deposited phosphate coatings, smooths out irregularities. When phosphate coatings are deposited from solutions with calcium nitrate or zinc additives, the phase content of white minerals (hopeite, parashcolzite) increases in their structural and phase composition, which cause the color change of the coating.
3.  White phosphate coatings obtained at room temperature are inferior in protective properties to unpainted, modified phosphate films. The corrosion rate of St3 steel samples protected by white phosphate coatings in a 3% NaCl solution is 3 times higher than that of samples with unpainted phosphate films. However, in comparison with unprotected steel samples, the corrosion rate of samples with white phosphate coatings is at a low level. This allows us to recommend the developed modified solutions of cold phosphating for the deposition of white phosphate coatings with good protective and decorative properties.

**Author Contributions:** Data curation, V.R. and V.K.; Formal analysis, V.R.; Investigation, V.K.; Methodology, V.K.; Project administration, E.R.; Resources, V.R.; Supervision, E.R.; Validation, E.R.; Writing—original draft, V.K.; Writing—review and editing, V.R. All authors have read and agreed to the published version of the manuscript.

**Funding:** This research received no external funding.

**Institutional Review Board Statement:** Not applicable.

**Informed Consent Statement:** Not applicable.

**Data Availability Statement:** Data sharing is not applicable to this article.

**Conflicts of Interest:** The authors declare no conflict of interest.

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
