# Peer review of "White Phosphate Coatings Obtained on Steel from Modified Cold Phosphating Solutions"

_coatings, doi:10.3390/coatings12010070_

Round 1

Reviewer 1 Report

In the manuscript entitled “White Phosphate Coatings Obtained on Steel from Modified Cold Phosphating Solutions”, the authors developed a method for obtaining white phosphate coatings on steel by cold method.

This paper presents a systematic study with detailed characterizations and analyses. However, some areas listed below need to be addressed. Therefore, I suggest major revision.

1. Materials and methods

1.1 The working mechanism of XRD is textbook knowledge. Thus, it is redundant to be included in a research paper.

2. Results

2.1. How the grain size is determined should be discussed. The selection of grains in Fig.3 is random and not in a systematic way.

3. Discussion

3.1 The XRD patterns of some phases of the coating in Fig. 9-11 are buried in the noise. It is not convincing for phase identification.

3.2 In line 184, it is claimed that “Since the smaller grain size causes higher performance properties of coatings”. The mechanism behind it should be discussed.

3.3 In line 206, it is concluded that “but their protective ability can be increased by additional coating with a colorless varnish”. This is supported by any evidence or discussed in the manuscript.

Author Response

We thank the reviewer for his clear comments and recommendations to the article.
The following changes have been made to the article:

1.1. section 2.3. X-ray analysis of the phase composition of phosphate coatings has been shortened;

2.1. Figure 3 shows sample data on grain size changes on the surface of phosphate coatings. An explanation has been added to section 2.2.: "To determine the average grain size of the deposited phosphate coatings, a grid was constructed on the intermediate divisions on the axes in the images. The grain diameter was measured at the intersections of the lines.";

3.1. noise has been removed from Figures 9-11;

3.2. regarding the phrase “Since the smaller grain size causes higher performance properties of coatings” an explanation is given: "Since the smaller grain size causes higher performance properties of coatings [32–35], it is obvious that the introduction of zinc or calcium nitrate additives into phosphating solutions adversely affects the protective properties of the deposited phosphate films. Fine-crystalline phosphate coatings have high adhesion to the metal surface and provide an effective physical barrier to protect corroded metal products from the environment. Fine-crystalline coatings have less porosity and fewer cracks and voids in their structure [36–39]. Due to their insulating nature, phosphate coatings prevent the occurrence and spread of corrosion.";

3.3. increasing the protective properties of phosphate coatings by applying varnish is a common practice, and it was suggested as a recommendation. Since studies of additional processing of phosphate coatings do not relate to the subject of this article, the phrase has been removed from Conclusion 3.

Reviewer 2 Report

I have reviewed the manuscript “White Phosphate Coatings Obtained on Steel from Modified Cold Phosphating Solutions” submitted to “Coatings” for publication. This is a well designed and well-conducted study in which authors have presented a method for obtaining white phosphate coatings on steel using a cold method. The manuscript fits well within the scope of the journal; it needs some major improvements; there are a few suggestions that authors may consider improving it further:

The use of English language is reasonable, however, there are a number of minor punctuation and grammatical errors; that should be corrected and rephrased using academic English for a better flow of text for reader.

Abstract: is unstructured however, lack information about the results/finding of the study. Some of the key result can be added to the abstract section.

Introduction; is covering the background information and the rationale of the study effectively. It is suggested to add some further information in context of applications for such coatings and methods used.

Line 55: The surface of the samples is presanded, degreased and etched”.. please add details in brief.

Eq. (1,2, and 3) please cite the source of this?

Figure 2: how these images were takes? What key features are shown?

Why figure 9-12 are part of discussion instead of results?

The discussion is very weak and authors should expand this section including further previous studies in the context.

Some of the statements from conclusion can be moved to the discussion section for further discussion.

Author Response

We thank the reviewer for his clear comments and recommendations to the article.
The following changes have been made to the article:

  • the values of the corrosion rate index to compare the protective properties of white and unpainted phosphate coatings have been added to the Abstract;
  • information about the methods of obtaining and applications of phosphate coatings has been added to the Introduction;
  • in section 2.1. Materials, an explanation of how the surface was prepared for phosphating has been introduced;
  • a link to the source for the formulas used is provided;
  • the caption to Figure 2 has been changed: Images of the surface of phosphated steel samples obtained by a digital camera: (a) Unpainted phosphate coating from a modified solution; (b) White phosphate coating from a solution with the addition of zinc nitrate; (c) White phosphate coating from a solution with the addition of calcium nitrate;
  • Figures 9-11 from the Discussion section have been moved to the Results section;
  • changes have been made to the Discussion and Conclusion sections.

Round 2

Reviewer 1 Report

The authors have properly addressed the issues. I recommend acceptance for publication in present form.

Reviewer 2 Report

Many thanks for the revision